# The Utility of Blood Cultures in Non-Febrile Patients and Patients with Antibiotics Therapy in Internal Medicine Departments

**DOI:** 10.3390/jcm14072373

**Published:** 2025-03-30

**Authors:** Yaniv Cojocaru, Lior Hassan, Lior Nesher, Tali Shafat, Victor Novack

**Affiliations:** 1Clinical Research Center and Division of Internal Medicine, Soroka University Medical Center, Faculty of Health Sciences, Ben-Gurion University of the Negev, Beer-Sheva P.O. Box 84101, Israel; cojocaru.yaniv@gmail.com (Y.C.); lior0351@gmail.com (L.H.); 2li.shafat@gmail.com (T.S.); 2Infectious Diseases Institute, Soroka University Medical Center, Faculty of Health Sciences, Ben-Gurion University of the Negev, Beer-Sheva P.O. Box 84101, Israel; nesherke@bgu.ac.il

**Keywords:** utility, blood cultures, non-febrile, antibiotics, optimal timing, bacteriological yield, bacteremia

## Abstract

**Background**: The injudicious use of blood cultures is associated with low cost-effectiveness and leads to unnecessary follow-up tests for false-positive results. In addition, false negatives can result in missed diagnoses, leading to delays in initiating appropriate treatment and potentially worsening patient outcomes. The timing of the blood culture tests related to the highest diagnostic yield is not fully elucidated. We hypothesized that a high proportion of the tests are done within non-optimal timing, resulting in a lower clinical yield. We specifically focused on the consequences of BC obtained in afebrile patients. **Methods**: We assessed 73,787 blood cultures taken between 2014 and 2020 in patients hospitalized with a suspected infection. Blood cultures were considered taken at optimal timing if the per rectum temperature was 38.3 °C or more and no prior antibiotics were given. Only the first culture per patient was assessed. The primary outcome was a true bacteremia defined by the clinically important pathogen. **Results**: Therefore, 25,616 blood cultures were obtained at optimal timing (34.7%), with true bacteremia found in 6.15% vs. 5.15% in cultures obtained at non-optimal timing. In a multivariable model, optimal timing adjusted for the variety of the clinical, demographic, and laboratory findings’ optimal timing was significantly associated with an increase in the odds of detecting true bacteremia (OR:1.23, 95% CI: 1.12–1.35). **Conclusions**: Nearly two-thirds of patients hospitalized due to a suspected infection did not have their blood cultures taken at the optimal time. Our findings underscore the importance of integrating clinical judgment, patient-specific risk factors, and evidence-based criteria when deciding to perform blood cultures, rather than relying solely on fever as an indicator.

## 1. Background

Bloodstream infections are an important cause of morbidity and mortality. The most common indications for obtaining blood cultures are the new onset of persistent fever of more than 38 degrees Celsius, leukocytosis, absolute granulocytopenia, or combinations of these clinical parameters [1,2]. Yet, several studies conducted over the years have demonstrated a lack of correlation between a patient’s clinical parameters and the risk for bacteremia [2,3,4].

While established guidelines delineate the requisite volume of blood and the number of blood cultures to be procured, ongoing discourse persists concerning the indications and timing thereof [5,6,7]. Despite the well-known clinical paradigm of obtaining blood cultures close to a fever spike, only a handful of studies have been conducted to validate this assumption. The study by Tabriz et al. (2004) [1] evaluated the practice patterns of repeating blood cultures during hospital stays and proposed guidelines for their use. This study involved a retrospective review of 1436 blood culture sets obtained from 1000 patients over a one-year period. The results indicated that repeat blood cultures were frequently ordered without clear clinical indications. Specifically, 50% of repeat cultures were performed within 24 h of the initial positive culture and 75% within 48 h. The study found that only 20% of repeat cultures yielded the same organism as the initial culture, suggesting that many repeat cultures may be unnecessary [1]. Riedel et al. (2008) [3] investigated the timing of blood culture specimen collection in febrile patients with bacteremia. Their results indicated that the timing of blood culture collection significantly affects the yield of positive results. Blood cultures collected before the administration of antibiotics had a higher positivity rate compared to those collected after antibiotic administration. Specifically, pre-antimicrobial blood cultures were positive in 31.4% of cases, whereas post-antimicrobial cultures were positive in only 19.4% of cases, with a statistically significant absolute difference of 12.0% (95% CI, 5.4–18.6%; *p* < 0.001) [3]. A study in pediatric patients showed that fever documented antecedent to blood culture procurement lacked sensitivity and specificity in predicting a positive culture outcome. However, it was demonstrated that pre-administered antibiotics are associated with a diminished likelihood of yielding positive cultures, thus corroborating analogous findings in adult cohorts [4].

We hypothesized that, despite the well-recognized clinical recommendation for blood culture test timing, a high proportion of the tests are done within the less-than-optimal timing defined as lack of fever and/or after the administration of the antibiotics, resulting in a lower clinical yield.

## 2. Methodology

### 2.1. Search Strategy

A comprehensive literature search was conducted to identify relevant studies that investigated the relationship between bacteremia, blood culture results, and the effectiveness of diagnostic techniques and antibacterial agents. The following MeSH terms were utilized: Bacteremia, Blood Culture, Time Factors, Sensitivity and Specificity, Sepsis, Anti-Bacterial Agents, and Diagnostic Techniques and Procedures.

### 2.2. Study Design

We initiated the study by screening blood culture samples collected from hospitalized patients in the internal medicine departments at Soroka University Medical Center (SUMC) from 2014 to 2020. Subsequently, we performed an additional screening to identify and exclude duplicate and repetitive blood culture samples, ensuring a dataset suitable for statistical analysis (Figure 1).

### 2.3. Study Settings

The study was carried out at the SUMC, a 1200-bed tertiary teaching hospital in Southern Israel, directly serving a population of 750,000. It is the second-largest hospital in Israel and serves as a referral center for more than 1.2 million people. The study included first blood cultures in all the patients admitted to the internal medicine departments in SUMC (260 beds) who had blood cultures tested. In our institute, blood cultures are obtained as one set of two bottles (one aerobe culture and one anaerobe culture), and the recommended volume is 9 mL of blood. We included only patients admitted directly to internal medicine wards from the emergency department, excluding those who required repetitive blood cultures as part of treatment monitoring (e.g., infective endocarditis), rather than as part of the initial fever workup. Cases of patients transferred to the internal medicine wards from surgical wards were not taken as part of the study due to the fact that the chances of bacteremia is higher in those patients because of a diagnosed infectious focus. All required data for this study were collected from SUMC computerized medical records, including demographics and clinical and laboratory characteristics.

### 2.4. Definitions

We stratified the blood culture results into three groups. The first group included true-positive blood cultures with clinically significant microorganisms, such as *Staphylococcus aureus*, *Streptococcus pneumoniae*, *Group A Streptococcus (GAS)*, *Enterobacteriaceae*, *Haemophilus influenzae*, *Pseudomonas aeruginosa*, *Bacteroidaceae*, *and Candida species*. The second group comprised patients with positive blood culture d/t microorganisms commonly associated with contamination, including *coagulase-negative staphylococci*, *Corynebacterium species*, *Cutibacterium*, *Bacillus species*, and *Micrococcus species*. When pathogens typically associated with contamination were present in more than two sets of blood cultures, we excluded them from the analysis, as they might indicate a clinical syndrome requiring follow-up with repeated blood culture testing.

Optimal timing for blood culture collection was defined based on two specific criteria: documented fever and the absence of prior antibiotic administration. Fever was identified using one of the two validated methods employed at our institution—rectal or oral temperature measurement—with a threshold of ≥38.3 °C recorded within 12 h before or after blood culture collection, accounting for potential delays in sample transport to the microbiology laboratory. Although rectal temperature measurement is an invasive procedure that may raise concerns regarding patient comfort and dignity, it is well documented that rectal measurements provide greater sensitivity for detecting fever compared to non-invasive methods, which may underestimate the core body temperature and fail to identify febrile states. The second condition defining optimal timing was the collection of blood cultures before the administration of any antibiotic therapy [8,9,10,11].

### 2.5. Statistical Analysis

We presented categorical variables as counts and percentages, analyzed by the chi-squared test, and calculated the Standardized Mean Difference (SMD). We developed the first multivariable logistic regression model to identify independent factors associated with optimal timing blood culture testing. The second logistic regression model was constructed to explore the relationship between optimal timing blood cultures and diagnostic yield (true-positive blood culture results), with adjustments made for various covariates. Clinical relevance and statistical significance (*p* < 0.10 in the univariable analysis) were considered to select the variables to be included in the models. A sensitivity analysis was conducted in which the absence of prior antibiotics, the presence of fever, and their interaction were introduced into the model as separate exposures. This was compared to a single exposure comprising both the presence of fever and the absence of prior antibiotic administration, allowing us to rule out potential data obstruction. The variance inflation factor (VIF) value assessed collinearity between the variables, with values above 5 suggesting at least a moderate degree of collinearity. All statistical analyses were performed using IBM SPSS, version 23.0, and a *p*-value of <0.05 was used as a significance threshold.

## 3. Results

From 2014 to 2020, 73,787 blood cultures were taken in the internal medicine department admission setting. Of this, 34.7% were acquired around an episode of fever and before initiating the antibiotics, while the remaining 65.3% were obtained not optimally. Of the 48,174 (65.3%) samples that were taken at non-optimal timing, 99% of the cases did not have a fever, and 2.4% already received antibiotics.

### 3.1. Patient Population

Table 1 presents patient population characteristics stratified by the timing of blood culture tests. The comparison between the optimal timing (*n* = 25,616) and not optimal timing (n = 48,171) groups reveals that most variables exhibit minor differences, as indicated by their Standardized Mean Difference (SMD) of less than 0.10. Vital signs show some variability, with a moderate difference in pulse rate (SMD = 0.458), indicating a higher pulse rate in the optimal timing group. In contrast, systolic blood pressure differences remain minimal (SMD = 0.083). The blood test results reflected minor differences, including white blood cell count (SMD = 0.159) and C-reactive protein levels (SMD = 0.133).

### 3.2. Bacteriological Results

Table 2 represents the association between the timing of obtaining blood cultures and the bacteriological results. True bacteremia was more common in the optimal timing group (6.15% vs. 5.15%, *p* < 0.001), and contamination rates were higher in the non-optimal timing group (2.53% vs. 2.38%, *p* < 0.001).

### 3.3. Multivariable Analysis

Table 3 demonstrates several characteristics associated with an increased likelihood of obtaining blood cultures during optimal timing in a multivariable logistic regression analysis. An increase in systolic blood pressure by 10 mmHg corresponded to an increase in the odds (OR: 1.06, 95% CI: 1.04–1.06). The presence of dementia (OR: 1.23, 95% CI: 1.14–1.34) and hemiplegia or paraplegia (OR: 1.30, 95% CI: 1.22–1.40) were associated with the suboptimal timing as well.

Table 4 depicts several clinical factors that exert statistically significant positive effects on the detection of true bacteremia: optimal timing was significantly associated with an increase in the odds of detecting true bacteremia (OR: 1.23, 95% CI: 1.12–1.35). While in term of patients’ comorbidities, diabetes mellitus with chronic complications significantly increased the odds (OR: 1.38, 95% CI: 1.22–1.44), moderate-to-severe liver disease was associated with a substantial increase in the odds (OR: 2.21, 95% CI: 1.67–2.93). The presence of any malignancy was associated with higher odds (OR: 1.20, 95% CI: 1.07–1.33). Lastly, AIDS was associated with a significant increase in the odds (OR: 2.32, 95% CI: 1.32–4.05). We have conducted a sensitivity analysis in which no prior antibiotics or presence of fever, together with the interaction between them, were introduced into the model as separate exposures. This analysis revealed that, while the interaction was not associated with a higher chance of obtaining a true-positive culture, both exposures were significant: no prior antibiotics therapy OR at 1.86 (95% CI 1.45–2.40) and presence of fever at 1.26 (95% CI 1.15–1.38).

Table 5 demonstrates that several clinical factors exert statistically significant positive influences on the likelihood of contamination. Optimal timing was associated with an increase in the odds (OR: 1.17, 95% CI: 1.03–1.34). The presence of a cerebrovascular accident (CVA) was linked to an increase in the likelihood of contamination (OR: 1.25, 95% CI: 1.07–1.45).

Appendix A compares patients with true bacteremia (n = 4045, 5.5%) and those with contamination and growth (n = 69,745, 94.5%) across various characteristics. Most characteristics show negligible to small differences between patients with true bacteremia and those with contamination and growth, and moderate differences are observed in the vital signs and blood test parameters. Specifically, there are moderate differences in the pulse rate (SMD = 0.154) and systolic blood pressure (SMD = 0.258), as well as in the white blood cell count (SMD = 0.240) and C-reactive protein levels (SMD = 0.239). Additionally, renal disease shows a small-to-moderate difference (SMD = 0.104).

## 4. Discussion

In this retrospective study spanning the years 2012–2020 and involving 73,787 blood cultures collected in the internal medicine departments of the second-largest hospital in Israel, we have elucidated several important clinical points. It appears that two-thirds of the blood cultures were obtained under non-optimal conditions, such as when the patient was either non-febrile or had already initiated antibiotic therapy. As hypothesized, obtaining blood cultures from febrile patients before initiating therapy was associated with a 23% increase in true bacteremia (absolute difference of 1%) [OR: 1.23, CI: 1.12–1.35, *p* < 0.001]. Yet, it must be emphasized that the proportion of the true-positive cultures was above 5%, even in patients whose blood cultures were obtained in an afebrile state or after initiating antibiotic therapy; however, the difference was small, and the clinical relevance still remains questionable.

The primary reasons for suboptimal blood culture timing were either the absence of a documented fever or the collection of cultures when the fever was present but patients were already receiving antibiotic treatment. Remarkably, our study revealed that approximately 99% of cultures taken at suboptimal times lacked fever documentation, prompting inquiry into the rationale guiding physicians’ sampling decisions. Prior research from 1997 has indicated that the optimal timing for blood culture collection is characterized by a sharp fever escalation in the absence of antibiotic treatment [12]. Conversely, studies conducted in adult and pediatric populations have suggested that fever alone may not enhance the likelihood of positive blood culture results indicative of bacteremia. Furthermore, it should be noted that guidelines, such as those from UpToDate, recommend obtaining blood cultures from patients with suspected syndromes closely associated with bacteremia [13]. In 2024, the Infectious Diseases Society of America (IDSA) and the American Society for Microbiology (ASM) published an update to the guidelines, which states that the most important variable in recovering bacteria and fungi from patients with bloodstream infections is the volume of blood that should be in adults (20–30 mL of blood per cultures) [14]. These diverse perspectives on optimal blood culture timing elucidate our findings, indicating a prevailing preference among our institution’s physicians to collect blood cultures upon fever onset under the assumption of increased bacteremia diagnostic yield.

To explore additional factors influencing the decision to obtain blood cultures for febrile patients, we developed a model examining the impact of underlying diseases and clinical indicators beyond fever. Our findings demonstrate that conditions linked to immunosuppression, such as malignancies and those predisposing patients to infections, significantly elevate the likelihood of blood culture necessity in febrile patients and the probability of detecting true bacteremia. Despite previous investigations into the ideal patient criteria for blood culture collection, the influence of underlying diseases on this decision remains largely unexplored [15,16].

Our study underscores the heightened likelihood of detecting true bacteremia when blood cultures are obtained during fever episodes without preceding antibiotic administration. Conversely, suboptimal timing of blood culture collection was associated with true bacteremia in 5% of cases, highlighting its relevance as a contributing factor. This finding aligns with a pediatric study demonstrating that 95% of children with positive blood cultures and 94% with negative cultures exhibited a febrile event within 12 h surrounding the blood collection time [3]. Similar trends were observed in adult populations [4].

The strengths of our study include the robustness of our databases, which enable comprehensive comparisons of blood culture yields and various factors beyond timing alone. Several limitations were encountered in our research, notably its retrospective nature and single-center design, potentially limiting generalizability across different medical settings. The exact timing of the blood drawing vis-à-vis the temperature measurement could not be estimated with a narrower than 12-h window resolution. This limitation can lead to a bias toward zero in the assessment of the hypothesis.

Another limitation that can be associated with the misclassification bias is the lack of information on whether the patient had a fever at home or was treated with antibiotics at home by the primary care physician. The exclusion of the patients with suspected contaminants found in more than one blood culture could potentially lead to a misclassification bias, yet the low number of such cases (less than 20) reassures that the study results are robust. Our results should not be generalized toward the population with a suspected surgical infection, as these patients were excluded from the present analysis.

Antibiotic treatment can be classified into three primary types: prophylactic, empirical, and therapeutic. Therapeutic treatment targets specific infections based on confirmed microbiological evidence. Empirical treatment, the focus of this study, is initiated when an infection is strongly suspected but microbiological confirmation is not yet available, making it particularly vital in critically ill patients where treatment delays can have severe consequences. Our study examined patients receiving empirical antibiotic therapy blood culture collection. Fever in this context is clinically significant, often indicating potential treatment failure, secondary infections, or even non-infectious causes. We observed that obtaining blood cultures during ongoing antibiotic therapy significantly reduced the likelihood of isolating clinically significant bacteremia, likely due to prior antibiotic exposure reducing the bacterial load and increasing the risk of false-negative results. However, despite this limitation, empirical therapy remains a cornerstone of infectious disease management. Even in cases of negative blood cultures, it is justifiable when guided by clinical judgment, local antibiograms, and knowledge of microbial profiles within the healthcare setting. Our findings revealed comparable clinical outcomes between culture-negative and culture-positive patients managed with empirical therapy, further supporting its utility. Our findings demonstrate that conditions linked to immunosuppression, such as malignancies and those predisposing patients to infections, significantly elevate the likelihood of blood culture necessity in febrile patients and the probability of detecting true bacteremia. Despite previous investigations into the ideal patient criteria for blood culture collection, the influence of underlying diseases on this decision remains largely unexplored. To optimize antibiotic use and mitigate the risks of antimicrobial resistance and misuse, antibiotic stewardship programs are crucial. These programs enhance clinical governance by promoting evidence-based, targeted, and sustainable antimicrobial practices, ultimately improving patient outcomes and preserving the efficacy of existing antibiotics for future use. Together, these findings underscore the importance of combining empirical antibiotic therapy with rigorous diagnostic, clinical, and stewardship efforts to achieve optimal infection management and sustainable antimicrobial use [17,18].

In conclusion, our findings underscore the importance of integrating clinical judgment, patient-specific risk factors, and evidence-based criteria when deciding to perform blood cultures, rather than relying solely on fever as an indicator. This approach may improve diagnostic accuracy and support more effective and targeted patient management, aligning with the principles of antibiotic stewardship and resource optimization.

## Figures and Tables

**Figure 1 jcm-14-02373-f001:**
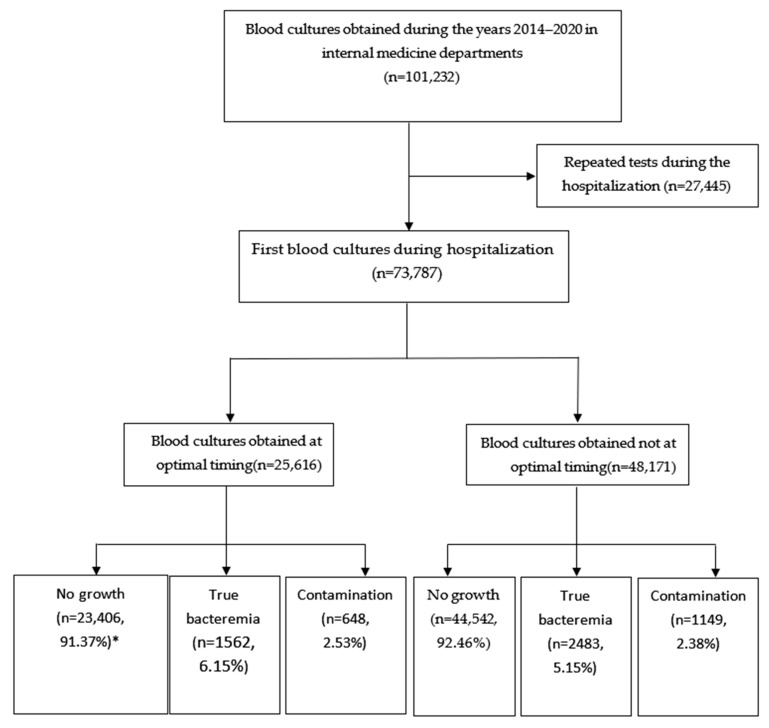
Study flow. * Percentage from the respective timing group (optimal and not optimal).

**Table 1 jcm-14-02373-t001:** Patient characteristics divided into groups according to the timing of the blood culture testing.

Characteristics	Optimal Timing, n = 25,616 (34.7%)	Non Optimal Timing, n = 48,171 (65.3%)	SMD
Age (y)	64.4 ± 18.4	65.7 ± 18.1	0.052
Gender			
Male, n (%)	14,804 (57.8%)	28,600 (59.4%)	0.031
Comorbidities			
CHF*, n (%)	4775 (18.64%)	11,893 (24.68%)	0.152
Stroke, n (%)	6286 (24.54%)	11,906 (24.71%)	0.067
Dementia, n (%)	2011 (0.8%)	2854 (5.92%)	0.013
Chronic pulmonary disease, n (%)	6048 (23.61%)	13,674 (28.4%)	0.071
Rheumatological disease, n (%)	886 (3.46%)	1972 (4.1%)	0.079
Diabetes mellitus, n (%)	9665 (37.73%)	19,199 (3.96%)	0.066
Diabetes mellitus with chronic complications, n (%)	3535 (13.8%)	7326 (15.2%)	0.032
Peripheral Vascular disease, n (%)	3540 (13.82%)	7820 (16.23%)	0.041
Malignancy, n (%)	4768 (18.6%)	9365 (19.44%)	0.132
Renal disease, n (%)	5966 (23.29%)	14,166 (29.4%)	0.011
Vital signs			
Pulse (bpm*)	100.7 ± 26.4	90.2 ± 19.6	0.458
Systolic Blood pressure (mmHg*)	134.1 ± 22.5	132.2 ± 23.0	0.083
Blood tests			
White blood cells (mm^3^/mL*)	11.39 ± 7.2	12.24 ± 9.5	0.159
C-reactive protein (mg/dL*)	15.29 ± 13.78	12.72 ± 11.19	0.133
Primary diagnosis			
Symptoms, Signs, And Ill-Defined Conditions, n (%)	5376 (20.99%)	10,344 (21.47%)	0.279
Diseases of Respiratory system, n (%)	3959 (15.46%)	7000 (14.53%)	0.074
Diseases of Circulatory system, n (%)	3739 (14.6%)	7122 (14.78)	0.042
Infectious and Parasitic Diseases, n (%)	3581 (13.98%)	6927 (14.38%)	0.010
Diseases of Genitourinary system, n (%)	1490 (5.82%)	2799 (5.81%)	0.035
Others, n (%)	1345 (5.25%)	2413 (5.01%)	0.068
Injury and Poisoning, n (%)	1291 (5.04%)	2446 (5.08%)	
Diseases of Digestive system, n (%)	1133 (4.42%)	2416 (5.02%)	0.054
Diseases of Skin and subcutaneous tissue, n (%)	794 (3.10%)	1329 (2.76%)	
Endocrine, Nutritional and Metabolic Diseases, And Immunity Disorders, n (%)	730 (2.85%)	1331 (2.76%)	0.033
Diseases musculoskeletal and connective tissue, n (%)	693 (2.71%)	1269 (2.63%)	0.012
Diseases of CNS, n (%)	625 (2.44%)	1190 (2.47%)	0.016
Neoplasms	558 (2.18%)	1027 (2.13%)	0.010
Diseases of blood and blood forming organ	302 (1.18%)	561 (1.16%)	0.008

* bpm—beats per minute, mg—milligram, dL—deciliter, mL—milliliter, mmHg—millimeter of mercury, and CHF—congestive heart failure.

**Table 2 jcm-14-02373-t002:** Association between timing of obtaining blood cultures and bacteriological yield and bacteriological results.

Bacteriological Yield	Optimal Timing, n = 25,616 (34.7%)	Not Optimal Timing, n = 48,174 (65.3%)	*p*-Value
No growth, n (%)	23,406 (91.37%)	44,542 (92.46%)	<0.001
Positive—true bacteremia *, n (%)	1562 (6.15%)	2483 (5.15%)	
Positive—contamination **, n (%)	648 (2.53%)	1149 (2.38%)	

* Clinically significant bacteria—*S. aureus*, *Streptococcus pneumoniae*, GAS, *Enterobacteriaceae*, *Haemophilus influenza*, *Pseudomonas aeruginosa*, *Bacteroidaceae*, and *Candida* species. ** Bacteria associated with contamination—Coagulase-negative Staphylococci, *Corynebacterium* species, *Cutibacterium*, *Bacillus* species, and *Micrococcus* species.

**Table 3 jcm-14-02373-t003:** Factors associated with the optimal timing of the blood culture collection.

Characteristic	Odds Ratio	95% CI for Odds Ratio		*p*-Value
		Lower	Upper	
Age (years)	0.99	0.99	0.99	<0.001
Pulse (bpm), per 10 bpm	1.28	1.27	1.30	<0.001
Systolic Blood Pressure (mmHg), per 10 mmHg	1.06	1.05	1.07	<0.001
C-reactive protein (mg/dL) per 5 mg/dL	1.10	1.09	1.11	<0.001
White blood cells (10^6^/mL)	0.99	0.98	0.99	<0.001
Congestive heart failure	0.86	0.81	0.91	<0.001
Dementia	1.23	1.14	1.34	<0.001
COPD *	0.85	0.81	0.90	<0.001
Rheumatological disease	0.85	0.76	0.94	0.003
Diabetes Mellitus with chronic complications	0.93	0.88	0.97	<0.05
Hemiplegia/Paraplegia	1.30	1.22	1.40	<0.001
Renal disease	0.79	0.75	0.83	<0.001
Malignancy	0.92	0.87	0.98	0.007
Moderate to severe liver disease	0.79	0.66	0.95	0.013
AIDS *	0.53	0.36	0.77	0.001

* COPD—chronic obstructive pulmonary disease and AIDS—acquired immunodeficiency syndrome.

**Table 4 jcm-14-02373-t004:** Predictors of true bacteremia.

Characteristic	Odds Ratio	95% CI for the Odds Ratio		*p*-Value
		Lower	Upper	
Optimal timing	1.23	1.12	1.35	<0.001
Age (years)	1.01	1.00	1.01	<0.001
Pulse (bpm), per 10 bpm	1.05	1.04	1.07	<0.001
Systolic Blood Pressure (mmHg), per 10 mmHg	0.88	0.86	0.90	<0.001
C-reactive protein (mg/dL) per 5 mg/dL	1.06	1.04	1.07	0.043
White blood cells (10^6^/mL)	1.01	1.00	1.01	<0.001
Diabetes Mellitus with chronic complications	1.38	1.22	1.44	<0.001
Renal disease	1.15	1.04	1.28	0.006
Moderate to severe liver disease	2.21	1.67	2.93	0.001
Any Malignancy	1.20	1.07	1.33	<0.001
AIDS *	2.32	1.32	4.05	0.003

* AIDS—acquired immunodeficiency syndrome.

**Table 5 jcm-14-02373-t005:** Odds ratio for contamination.

Characteristic	Odds Ratio	95% CI for the Odds Ratio		*p*-Value
		Lower	Upper	
Optimal timing	1.17	1.03	1.34	0.018
Age (years)	1.01	1.00	1.01	<0.001
Gender—male	0.86	0.75	0.98	0.024
Systolic Blood Pressure (mmHg), per 10 mmHg	0.95	0.92	0.98	<0.001
C-reactive protein (mg/dL) per 5 mg/dL	0.96	0.93	0.99	0.024
CVA *	1.25	1.07	1.45	0.004
Dementia	0.76	0.58	0.99	0.041
COPD *	0.79	0.68	0.93	0.003
Renal disease	0.79	0.68	0.93	0.004
Rheumatological disease	0.48	0.30	0.79	0.003

* CVA—cerebrovascular accident and COPD—chronic obstructive pulmonary disease.

## Data Availability

The original contributions presented in this study are included in the article/Appendix A. Further inquiries can be directed to the corresponding author(s).

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
