# Peer review of "The Utility of Blood Cultures in Non-Febrile Patients and Patients with Antibiotics Therapy in Internal Medicine Departments"

_jcm, 2025, doi:10.3390/jcm14072373_

Round 1

Reviewer 1 Report

Comments and Suggestions for Authors

This is a very important topic and hence i read with interest the study. In my own experience, the blood cultures are positive in about 35% of patients who have severe sepsis and are treated with antibiotics or surgical source control i.e. about 2/3rd patient blood culture is negative despite patient having sepsis. This i consider due to timing of blood culture, sensitivity of current microbial detection techniques and sometimes patient already being started on antibiotics prior to cultures. I have some comments:

1. In abstract you mention the problem of false positives. To me, the main problem or the more common problem is false negatives rather than false positives. So, consider rephrasing the problem statement. 

2. I have some concerns with your definition of optimal timing of doing cultures. Firstly, this is a retrospective study, so this may not be consistent in all patients. I say so as even if hospital has a policy to do, the compliance is never 100%. Secondly, your range of 12 hours before or after fever is too wide and to me, not consistent with clinical practice of doing or advocating to do cultures during the spike. Plus minus couple of hours is understandable, but 12 hours is too long a time period to be defined as 'optimal timing". In fact, to me it is suboptimal timing! Finally, I am unsure why these day and age (2014 to 2020) we measure rectal temperatures when they can be done preserving patient dignity by many simpler means? 

3. Table 2 - No growth - is repeated twice. Why do you mention fermenting and non-fermenting bacteria? Possible to make it clinical description rather than microbiologic jargon that clinicians will find difficult to dechiper.

4. Table 1 shows dementia patients had blood cultures done at suboptimal timings while line 143-144 mentions that they had taken during optimal timings. Pls check and correct.

5. Table 3,4 and 5 is just multivariate data and ususally this should not be separate table but included with raw data table merged with multivariate OR and 95%CI columns. So see if the results can be presented in some simpler form with less tables and avoiding repetitive table headings/subheadings

6. Line 182-182 the mention of relative 23% increase can serve to mislead the readers with regards to the impact of the results/data and please also mention absolute increase in brackets so readers know that the actual difference is trivial and the p value is significant largely due to large sample size. 

7. In your institute, what is the volume of blood collected? How many bottles? 

8. You are absolutely right about using clinical judgment. In management, sometimes even if culture is negative, empiric use of antibiotic is permissible, and doctor is within his right to start a patient on empiric antibiotics based on clinical suspicion. We have shown that empiric antibiotic management of patients based on understanding of local antibiogram and local microbial profile, the outcomes of management of culture negative patients and culture positive patients are same - PMID: 27733320. You should discuss this too.

9. In clinical management, 3 different terminologies are important - prophylactic, empirical and therapeutic with clear definitions and implications in management and outcomes -. You should link your results and discuss these issues in patient care. 

10. There is good data that antibiotic stewardship teams are important in monitoring and governance of antibiotic use and prevent misuse/abuse. I do not see any mention of antibiotic stewardship and please include some elements in relevant parts in discussion segment -  .

Thanks

Author Response

Response to reviewer 1:

  1. In abstract you mention the problem of false positives. To me, the main problem or the more common problem is false negatives rather than false positives. So, consider rephrasing the problem statement. 
  • We agreed with your statement and therefore we have rephrased it in the manuscript: "The injudicious use of blood cultures is associated with low cost-effectiveness and leads to unnecessary follow-up tests for false positive results. In addition, false negatives can result in missed diagnoses, leading to delays in initiating appropriate treatment and potentially worsening patient outcomes. The timing of the blood culture tests related to the highest diagnostic yield is not fully elucidated".
  1. I have some concerns with your definition of optimal timing of doing cultures. Firstly, this is a retrospective study, so this may not be consistent in all patients. I say so as even if hospital has a policy to do, the compliance is never 100%. Secondly, your range of 12 hours before or after fever is too wide and to me, not consistent with clinical practice of doing or advocating to do cultures during the spike. Plus minus couple of hours is understandable, but 12 hours is too long a time period to be defined as 'optimal timing". In fact, to me it is suboptimal timing! Finally, I am unsure why these day and age (2014 to 2020) we measure rectal temperatures when they can be done preserving patient dignity by many simpler means? 
  • We agree with the notion that the compliance with any rule in the field of medicine should not be expected to be 100%. In a way this low adherence to the recommendations is that very thing that allowed us to explore the question whether the recommendations (e.g. wait to fever to draw the cultures) are valid in the first place?
  • We concur with the reviewer that using the liberal window for the body temperature assessment to define the optimal timing (fever during ±12 hours) could lead to the misclassification bias where some non-optimal timing cultures (no fever) were defined as drawn at optimal timing. Unfortunately, the exact timing of the blood drawing vis-à-vis the temperature measurement could not be estimated with a narrower than 12-hour window resolution. This is certainly the limitation of the study and can lead to the bias toward zero in the assessment of the hypothesis. We have addressed this limitation in the discussion: " The exact timing of the blood drawing vis-à-vis the temperature measurement could not be estimated with a narrower than 12-hour window resolution. This limitation can lead to the bias toward zero in the assessment of the hypothesis."
  • We agree with the reviewer that in the majority of the clinical situations obtaining rectal temperature can be unnecessary invasive. Yet, in the hospital setting measuring temperature by any non-invasive method is not as reliable as a rectal temperature for detecting fever. Furthermore, numerous clinical studies attempt to show correlations between oral and rectal temperatures, but ultimately demonstrated reduced sensitivity to identify fever with non-rectal measurements in a number of clinical scenarios.[9-10]
  1. Table 2 - No growth - is repeated twice. Why do you mention fermenting and non-fermenting bacteria? Possible to make it clinical description rather than microbiologic jargon that clinicians will find difficult to decipher.

  • We thank the reviewer for insight and therefore we decided to remove the terms of fermenting and non- fermenting bacterias from table number 2.
  1. Table 1 shows dementia patients had blood cultures done at suboptimal timings while line 143-144 mentions that they had taken during optimal timings. Pls check and correct.
  • We thank the reviewer for the highlight of mistake and now the text in the manuscript is "The presence of dementia (OR: 1.23, 95% CI: 1.14-1.34) and hemiplegia or paraplegia (OR: 1.30, 95% CI: 1.22-1.40) were associated with the sub-optimal timing as well."
  1. Table 3,4 and 5 is just multivariate data and usually this should not be separate table but included with raw data table merged with multivariate OR and 95%CI columns. So see if the results can be presented in some simpler form with less tables and avoiding repetitive table headings/subheadings
  • We appreciate your suggestion and following it we have updated the titles of the tables but we agreed that mixing all the tables together will create inappropriate way to present our results.
  1. Line 182-182 the mention of relative 23% increase can serve to mislead the readers with regards to the impact of the results/data and please also mention absolute increase in brackets so readers know that the actual difference is trivial and the p value is significant largely due to large sample size. 
  • We have updated the text following your suggestion: "Obtaining blood cultures from febrile patients before initiating therapy was associated with a 23% increase in true bacteremia (absolute difference of 1%)"
  1. In your institute, what is the volume of blood collected? How many bottles? 
  • We added the sentence "In our institute, blood cultures are obtained as one set of two bottles (one aerobic culture and one anaerobic culture). Recommended volume is 9 ml of blood " in order to provide the future reader of the article the protocol of how to obtain blood cultures in our hospital. Unfortunately, we don’t have data regarding the volume of blood for each sample.
  1. You are absolutely right about using clinical judgment. In management, sometimes even if culture is negative, empiric use of antibiotic is permissible, and doctor is within his right to start a patient on empiric antibiotics based on clinical suspicion. We have shown that empiric antibiotic management of patients based on understanding of local antibiogram and local microbial profile, the outcomes of management of culture negative patients and culture positive patients are same - PMID: 27733320. You should discuss this too.
  2. In clinical management, 3 different terminologies are important - prophylactic, empirical and therapeutic with clear definitions and implications in management and outcomes -. You should link your results and discuss these issues in patient care. 
  3. There is good data that antibiotic stewardship teams are important in monitoring and governance of antibiotic use and prevent misuse/abuse. I do not see any mention of antibiotic stewardship and please include some elements in relevant parts in discussion segment.
  • We agree with your comments number 8-10 and following these comments we have updated the discussion section:

"Antibiotic treatment can be classified into three primary types: prophylactic, empirical, and therapeutic. Therapeutic treatment targets specific infections based on confirmed microbiological evidence. Empirical treatment, the focus of this study, is initiated when an infection is strongly suspected but microbiological confirmation is not yet available, making it particularly vital in critically ill patients where treatment delays can have severe consequences. Judicious use of the empirical antibiotics therapy is justifiable when guided by clinical judgment, local antibiograms, and knowledge of microbial profiles within the healthcare setting. To optimize antibiotic use and mitigate the risks of antimicrobial resistance and misuse, antibiotic stewardship programs are crucial especially in the case of the empirical therapy. These programs enhance clinical governance by promoting evidence-based, targeted, and sustainable antimicrobial practices, ultimately improving patient outcomes and preserving the efficacy of existing antibiotics for future use.

However, the ultimate goal of treating patient with the presumed bacterial infection remains the same - transition to the therapeutic antibiotics therapy (e.g. narrowing the spectrum) based on the cultures results. ”.[17-18]

Reviewer 2 Report

Comments and Suggestions for Authors

General comments

This is a retrospective audit that evaluated blood cultures taken between 2014 and 2020 in a hospital and concluded that nearly two-thirds of patients hospitalized due to the suspected infection did not have their blood cultures taken at the optimal time. Based on retrospective observation, this suboptimal timing reduced the likelihood of detecting true bacteremia compared to optimal time (5.15% vs. 6.15%) with statistical significance difference (OR:1.23, 95% CI: 26 1.12-1.35).

Some of the methodology is a bit ambiguous. For instance, what if coagulase-negative staph and “contaminants” show up in two sets of blood culture of the same patient – would that be considered contaminants? What about some other significant pathogens not included in this study list of pathogens (e.g. Viridans group streptococcus, obligate anaerobes other than Bacteroidaceae). Without knowing this detail, it is hard to determine whether this study had robust methodology.

The study define fever as per rectum temperature of 38.3 degrees Celsius or higher. But it is unclear what about measurement of fever by other means as such as forehead and axillary thermometers and measurement taken prior to arrival at the hospital. These patients would be classified to the group with “suboptimal blood culture timing.”

The data could be obscured when “suboptimal timing” can be either delay over 12 hours or prior antibiotics. As authors have referenced, prior antibiotics can reduce yield of blood culture. Then the lower yield of blood culture due to “suboptimal timing” could be mainly driven by these patients receiving prior antibiotics, rather than by delay over 12 hours. If possible, I would suggest separating delay over 12 hours and prior antibiotic groups into 2 separate categories.

It is difficult to determine the significance of this study. The authors may need to have more in depth background to demonstrate what the knowledge gap is, what the controversies are, and why their study is needed.

Although the authors show there were some clinical parameters leading to optimal blood culture timing, such as systolic blood pressure and pulse (Table 3). Many hospitals nowadays would have a sepsis protocol that advise blood culture taken when there are signs of fever with unstable vital signs. This manuscript needs better justifications on why the current study findings are novel.

Specific comments

Introduction

Line 38 and Line 43: It is a bit odd that the first references are reference 4-5. Please consider rearrange the numerical orders with your reference program.

Line 45-47: It may be important to clarify the data in Reference 5 – was the conclusion of “bacteremia detection probability remained unaltered” due to lack of sample size / statistical power? Also, this is a very old study done in 1989; the reference shows this is an abstract rather than a full article. Is there no better reference than this?

Line 47-54: Like the comment made regarding Reference 5, when you talk about negative results with no statistical significance in Reference 6 and 7, it may be important to briefly show the data.

Line 56-59: I think better justification is needed to explain the rationale for this study. You said “in this large cohort study.” Do you mean the previous studies failed to show difference due to small sample size, and that is why now you conducted a larger study? If that is what you meant, then, you probably need to go back to reference 5, 6, 7 to detail the sample size and statistical significance in each of these previous studies.

Methodology

Line 72-74: Some more explanations are needed on why surgical ward patients transferred to medicine are excluded. It is not shown in Figure 1 how many of these surgical patients were excluded. It is also very unclear what you mean by saying “patients in whom obtaining multiple blood cultures is a part of the diagnostic process.” Obtaining multiple blood cultures (which I interpret as obtaining more than 1 set of blood culture) is a common diagnostic procedure for suspected systemic infection.

Line 78-84: The list of microorganisms that can grow in blood culture go beyond the list of microorganisms in this paragraph. What about the rest of them? For instance, viridans group streptococcus commonly appear in blood culture. What about other obligate anaerobes that are not Bacteroidaceae, such as Clostridium, Peptostreptococcus, and Actinomyces? The definition of contaminant needs to be better defined. As per the American Society of Microbiology Clinical Microbiology Procedures Handbook and Clinical Laboratory Standard Institute M47 2nd edition, these “contaminants” would be considered as potential pathogens if they appear in both sets of blood culture. It is unclear how reference 14 (NHSN Terminology) help here.

Line 80: When you use abbreviations like GAS the first time, it is best to spell it out like Group A Streptococcus (GAS).

Line 85-89: What if the temperature was measured on axilla or forehead? Is rectal temperature a common practice in this hospital? What if patients refuse the rectal temperature measurement and opt for other measures?

The data could be obscured when “suboptimal timing” can be either delay over 12 hours or prior antibiotics. As you have referenced, prior antibiotics can reduce yield of blood culture. Then the lower yield of blood culture due to “suboptimal timing” could be mainly driven by these patients receiving prior antibiotics, rather than by delay over 12 hours.

Results

Line 134-135: For the readers not familiar with microbiology, it is best that you clarify which microorganisms were considered fermenting and non-fermenting gram-negative organisms in your methodology.

Line 147 Table 3: It is a bit odd that despite the odds ratio not crossing 1.0 for diabetes mellitus with chronic complications, the p-value did not reach statistical significance (p = 0.19). Are there explanations for that?

Line 168: I am not able to see the supplemental table 1 being attached and thereby cannot comment on that.

Discussion

Line 191: There could be other valid reasons why blood culture was ordered despite having no record of per rectum temperature of 38.3 degrees Celsius or higher in the hospital records.  Sepsis protocols could be in place that prompt clinicians to take blood culture even if patient does not present with fever e.g. patient with elevated inflammatory markers and unstable vital signs. Compared to the general population, geriatric patients are also less likely to spike fever. As stated in Clinical and Laboratory Standard Institute M47 2nd edition Section 3.1: “In the medical inpatient, fever alone is a poor predictor of bacteremia.”

Also, it is important to note that the clinical history influence whether a blood culture is needed. For instance, a patient can present with a subjective history of fever and chills prior to arrival to the emergency department. A patient can have objective measurement of fever in the general practice clinic or at home with an axillary or forehead thermometer prior to arrival to the emergency department. However, these patients may not necessarily have a recorded per rectum temperature of 38.3 degrees Celsius or higher in the hospital records. Depending on the clinical urgency, a physician may opt to obtain blood culture and then start an empiric antimicrobial therapy that would improve the fever for a systemic infection confirmed later with a blood culture result. But using the current study protocol, these incidents would be counted as “suboptimal” blood culture collecting time because there was no record of fever in hospital.

Should the current study data really lead to inquiry into the rationale guiding physicians' sampling decisions?

Line 233-236: The data collected does not seem to well support the conclusion. If your conclusion is about sensitivity and specificity of blood culture, then sensitivity and specificity data must be shown. The study never showed the pre-test probability of bacteremia. It is unclear what clinical suspicions and underlying diseases could lead to decision of blood culture orders.

References:

Line 267-268: Is reference 8 a journal article? If yes, please follow the proper format e.g. page number, year

Line 277-279: Is reference 13 a journal article? If yes, please follow the proper format e.g. page number

Author Response

Response to reviewer 2:

Some of the methodology is a bit ambiguous. For instance, what if coagulase-negative staph and “contaminants” show up in two sets of blood culture of the same patient – would that be considered contaminants? What about some other significant pathogens not included in this study list of pathogens (e.g. Viridans group streptococcus, obligate anaerobes other than Bacteroidaceae). Without knowing this detail, it is hard to determine whether this study had robust methodology.

  • We agree with the reviewer. Therefore, the methods section contains the following explanation: " We stratified the blood culture results into three groups. The first group included true-positive blood cultures with clinically significant microorganisms, such as Staphylococcus aureus, Streptococcus pneumoniae, Group A Streptococcus (GAS), Enterobacteriaceae, Haemophilus influenzae, Pseudomonas aeruginosa, Bacteroidaceae, and Candida species. The second group consisted of positive blood cultures containing microorganisms commonly associated with contamination, including coagulase-negative staphylococci, Corynebacterium species, Cutibacterium, Bacillus species, and Micrococcus species. However, some of these contaminants may act as potential pathogens.When pathogens typically associated with contamination were present in more than two sets of blood cultures, we excluded them from the analysis, as they might indicate a clinical syndrome requiring follow-up with repeated blood culture testing.
  • Furthermore, we have updated the limitation section to state: "The exclusion of the patients with suspected contaminant found in more than one blood culture could potentially lead to a misclassification bias."

The study define fever as per rectum temperature of 38.3 degrees Celsius or higher. But it is unclear what about measurement of fever by other means as such as forehead and axillary thermometers and measurement taken prior to arrival at the hospital. These patients would be classified to the group with “suboptimal blood culture timing.”

  • We agree with the reviewer comment regarding other means of measuring temperature and possible pre-hospital setting recorded temperature. However, per our study design we have classified the patient population based only on the oral or rectal temperature taken in the hospital close to the drawing of the blood cultures. The preadmission temperature was not recorded.

The data could be obscured when “suboptimal timing” can be either delay over 12 hours or prior antibiotics. As authors have referenced, prior antibiotics can reduce yield of blood culture. Then the lower yield of blood culture due to “suboptimal timing” could be mainly driven by these patients receiving prior antibiotics, rather than by delay over 12 hours. If possible, I would suggest separating delay over 12 hours and prior antibiotic groups into 2 separate categories.

  • We agree with the reviewer and have added a sensitivity analysis in which no prior antibiotics treatment and presence of fever are separated exposures. The results section was updated accordingly: "We have conducted a sensitivity analysis in which no prior antibiotics, presence of fever together with the interaction between them were introduced into the model as the separate exposures. This analysis revealed that while the interaction was not associated with a higher chance of obtaining a true positive culture, both exposures were significant: no prior antibiotics therapy OR of 1.86 (95% CI 1.45-2.40) and presence of fever 1.26 (95% CI 1.15-1.38)."

Although the authors show there were some clinical parameters leading to optimal blood culture timing, such as systolic blood pressure and pulse (Table 3). Many hospitals nowadays would have a sepsis protocol that advise blood culture taken when there are signs of fever with unstable vital signs. This manuscript needs better justifications on why the current study findings are novel.

  • We believe that the findings of the current study stress the need to relay on the holistic view of the signs and symptoms (and in a near future a personalized medicine approach), rather than follow strict protocols. We have shown that while the presence of fever and no prior antibiotics increase the chance of yielding a true positive culture, testing the blood cultures in patients with no fever and/or prior antibiotics therapy did result in the non-negligible rate of true bacteremia. Moreover, we have identified factors associated with the increase in the bacteriological yield that following the prospective validation may be used as a basis for the creation of the personalized testing protocols.

Specific comments

Introduction

Line 38 and Line 43: It is a bit odd that the first references are reference 4-5. Please consider rearrange the numerical orders with your reference program.

  • We have reorganized the references to ensure they follow the order of mention in the text.

Line 45-47: It may be important to clarify the data in Reference 5 – was the conclusion of “bacteremia detection probability remained unaltered” due to lack of sample size / statistical power? Also, this is a very old study done in 1989; the reference shows this is an abstract rather than a full article. Is there no better reference than this?

  • We appreciate the reviewer's feedback regarding Reference 5. The conclusion that "bacteremia detection probability remained unaltered" was influenced by the study’s limited sample size and statistical power. Additionally, the 1989 study (surprisingly widely cited) is an abstract rather than a full article, which emphasizes that relatively little is known about the exact timing of blood culture collection and the potential factors that may influence culture results. Our study is novel because there is a paucity of comprehensive data on the optimal timing for obtaining blood cultures. Existing literature primarily focuses on pre-analytical times and the impact of early versus late blood culture collection on yield but does not provide a detailed analysis of timing relative to clinical events. The novelty of our study lies in its in-depth examination of blood culture collection timing and its direct correlation with bacteremia detection rates. This approach fills a critical gap in current knowledge and provides evidence-based recommendations for optimizing blood culture practices.

In order to strengthen our claim, we outline the literature search strategy in our article as follows: "A comprehensive literature search was conducted to identify relevant studies that investigated the relationship between bacteremia, blood culture results, and the effectiveness of diagnostic techniques and antibacterial agents. The following MeSH terms were utilized: Bacteremia, Blood Culture, Time Factors, Sensitivity and Specificity, Sepsis, Anti-Bacterial Agents, and Diagnostic Techniques and Procedures." By addressing these key areas, our study contributes valuable insights into the timing of blood culture collection, ultimately aiming to improve diagnostic accuracy and patient outcomes.

Line 47-54: Like the comment made regarding Reference 5, when you talk about negative results with no statistical significance in Reference 6 and 7, it may be important to briefly show the data.

Line 56-59: I think better justification is needed to explain the rationale for this study. You said “in this large cohort study.” Do you mean the previous studies failed to show difference due to small sample size, and that is why now you conducted a larger study? If that is what you meant, then, you probably need to go back to reference 5, 6, 7 to detail the sample size and statistical significance in each of these previous studies.

  • We agree with your comments regarding lines 47–59 that more detail is required. Therefore, we have updated the background section with the following text: "Despite the well-known clinical paradigm of obtaining blood cultures close to a fever spike, only a handful of studies have been conducted to validate this assumption. The study by Tabriz et al. (2004) evaluated the practice patterns of repeating blood cultures during hospital stays and proposed guidelines for their use. This study involved a retrospective review of 1,436 blood culture sets obtained from 1,000 patients over a one-year period. The results indicated that repeat blood cultures were frequently ordered without clear clinical indications. Specifically, 50% of repeat cultures were performed within 24 hours of the initial positive culture, and 75% within 48 hours. The study found that only 20% of repeat cultures yielded the same organism as the initial culture, suggesting that many repeat cultures may be unnecessary.[1]Riedel et al. (2008) investigated the timing of blood culture specimen collection in febrile patients with bacteremia. Their results indicated that the timing of blood culture collection significantly affects the yield of positive results. Blood cultures collected before the administration of antibiotics had a higher positivity rate compared to those collected after antibiotic administration. Specifically, pre-antimicrobial blood cultures were positive in 31.4% of cases, whereas post-antimicrobial cultures were positive in only 19.4% of cases, with a statistically significant absolute difference of 12.0% (95% CI, 5.4% to 18.6%; P < 0.001)."

Given the small sample sizes of these studies, we decided to conduct a larger analysis with a larger sample size to validate our hypothesis.

  •  

Methodology

Line 72-74: Some more explanations are needed on why surgical ward patients transferred to medicine are excluded. It is not shown in Figure 1 how many of these surgical patients were excluded.

  • Per our design, we have focused our analysis on non-surgical infections. Therefore, we have excluded patients who were hospitalized in the surgical wards. We have updated our limitations section to state the following: "

 It is also very unclear what you mean by saying “patients in whom obtaining multiple blood cultures is a part of the diagnostic process.” Obtaining multiple blood cultures (which I interpret as obtaining more than 1 set of blood culture) is a common diagnostic procedure for suspected systemic infection.

  • We appreciate the reviewer comment and therefore we updated the methods section with this text: "We included only patients admitted directly to internal medicine wards from the emergency department, excluding those who required repetitive blood cultures as part of treatment monitoring )eg, Infective Endocarditis), rather than as part of the initial fever workup."

Line 78-84: The list of microorganisms that can grow in blood culture go beyond the list of microorganisms in this paragraph. What about the rest of them? For instance, Viridans group streptococcus commonly appear in blood culture. What about other obligate anaerobes that are not Bacteroidaceae, such as Clostridium, Peptostreptococcus, and Actinomyces? The definition of contaminant needs to be better defined. As per the American Society of Microbiology Clinical Microbiology Procedures Handbook and Clinical Laboratory Standard Institute M47 2nd edition, these “contaminants” would be considered as potential pathogens if they appear in both sets of blood culture. It is unclear how reference 14 (NHSN Terminology) help here.

  • We appreciate the comment. We have updated the "definitions" section as follow: "We stratified the blood culture results into three groups. The first group included true-positive blood cultures with clinically significant microorganisms, such as Staphylococcus aureus, Streptococcus pneumoniae, Group A Streptococcus (GAS), Enterobacteriaceae, Haemophilus influenzae, Pseudomonas aeruginosa, Bacteroidaceae, and Candida species. The second group consisted of positive blood cultures containing microorganisms commonly associated with contamination, including coagulase-negative staphylococci, Corynebacterium species, Cutibacterium, Bacillus species, and Micrococcus species. However, some of these contaminants may act as potential pathogens. When pathogens typically associated with contamination were present in more than two sets of blood cultures, we excluded them from the analysis, as they might indicate a clinical syndrome requiring follow-up with repeated blood culture testing."

Line 80: When you use abbreviations like GAS the first time, it is best to spell it out like Group A Streptococcus (GAS).

  • We corrected the text to: "Group A Streptococcus (GAS)"

Line 85-89: What if the temperature was measured on axilla or forehead? Is rectal temperature a common practice in this hospital? What if patients refuse the rectal temperature measurement and opt for other measures?

  • We agree with the reviewer that in the majority of the clinical situations obtaining rectal temperature can be unnecessary invasive. Yet, in the hospital setting measuring temperature by any non-invasive method is not as reliable as a rectal temperature for detecting fever. g. numerous medical textbooks and clinical studies attempted to provide correlations between oral and rectal temperatures, without much success [9-10]. Yet, the autonomy of the patient is important and if the patient refuse for rectal temperature we will use non-invasive way. In our institute the non-invasive way to measure temperature is per os.

The data could be obscured when “suboptimal timing” can be either delay over 12 hours or prior antibiotics. As you have referenced, prior antibiotics can reduce yield of blood culture. Then the lower yield of blood culture due to “suboptimal timing” could be mainly driven by these patients receiving prior antibiotics, rather than by delay over 12 hours.

  • We appreciate the reviewer's insight. Indeed, the data could be obscured when 'suboptimal timing' refers to either a delay in obtaining the cultures or delivering them after a spiking fever or antibiotic administration. Therefore, we have updated the statistical analysis section to clarify how we addressed this issue. "A sensitivity analysis in which the absence of prior antibiotics, the presence of fever, and their interaction were introduced into the model as separate exposures. This was compared to a single exposure comprising both the presence of fever and the absence of prior antibiotic administration, allowing us to rule out potential data obstruction."

Results

Line 134-135: For the readers not familiar with microbiology, it is best that you clarify which microorganisms were considered fermenting and non-fermenting gram-negative organisms in your methodology.

  • We agreed with your statement and removed those terms from the table and now the table appears differently

Line 147 Table 3: It is a bit odd that despite the odds ratio not crossing 1.0 for diabetes mellitus with chronic complications, the p-value did not reach statistical significance (p = 0.19). Are there explanations for that?

  • Thank you for pointing us to mistake in the text. After checking the data, the P value was <0.05.

Line 168: I am not able to see the supplemental table 1 being attached and thereby cannot comment on that.

  • Supplemental table 1 will be provided separately and not in the manuscript itself

Discussion

Line 191: There could be other valid reasons why blood culture was ordered despite having no record of per rectum temperature of 38.3 degrees Celsius or higher in the hospital records.  Sepsis protocols could be in place that prompt clinicians to take blood culture even if patient does not present with fever e.g. patient with elevated inflammatory markers and unstable vital signs. Compared to the general population, geriatric patients are also less likely to spike fever. As stated in Clinical and Laboratory Standard Institute M47 2nd edition Section 3.1: “In the medical inpatient, fever alone is a poor predictor of bacteremia.”

Also, it is important to note that the clinical history influence whether a blood culture is needed. For instance, a patient can present with a subjective history of fever and chills prior to arrival to the emergency department. A patient can have objective measurement of fever in the general practice clinic or at home with an axillary or forehead thermometer prior to arrival to the emergency department. However, these patients may not necessarily have a recorded per rectum temperature of 38.3 degrees Celsius or higher in the hospital records. Depending on the clinical urgency, a physician may opt to obtain blood culture and then start an empiric antimicrobial therapy that would improve the fever for a systemic infection confirmed later with a blood culture result. But using the current study protocol, these incidents would be counted as “suboptimal” blood culture collecting time because there was no record of fever in hospital.

Should the current study data really lead to inquiry into the rationale guiding physicians' sampling decisions?

  • We agree with the reviewer’s observation. Indeed, fever alone is a poor prognostic factor for identifying bacteremia in a patient. Therefore, in order to highlight the necessity of judicious use of blood cultures the discussion part has been updated with the follow: "Our findings demonstrate that conditions linked to immunosuppression, such as malignancies and those predisposing patients to infections, significantly elevate the likelihood of blood culture necessity in febrile patients and the probability of detecting true bacteremia. Despite previous investigations into the ideal patient criteria for blood culture collection, the influence of underlying diseases on this decision remains largely unexplored.
  • As the reviewer notes, patients may visit the emergency room after a fever is measured in a primary care clinic or at home. However, we chose to rely on temperature measurements taken during the triage phase, whether oral or rectal.

Line 233-236: The data collected does not seem to well support the conclusion. If your conclusion is about sensitivity and specificity of blood culture, then sensitivity and specificity data must be shown. The study never showed the pre-test probability of bacteremia. It is unclear what clinical suspicions and underlying diseases could lead to decision of blood culture orders.

  • We fully concur with the reviewer’s observation. Sensitivity, specificity, and, by extension, the positive and negative predictive values (PPV and NPV) of blood cultures were not the primary focus of our study. Therefore, our conclusion was changed as following : ""In conclusion, our findings underscore the importance of integrating clinical judgment, patient-specific risk factors, and evidence-based criteria when deciding to perform blood cultures, rather than relying solely on fever as an indicator. This approach not only improves diagnostic accuracy but also supports more effective and targeted patient management, aligning with the principles of antibiotic stewardship and resource optimization."

References:

Line 267-268: Is reference 8 a journal article? If yes, please follow the proper format e.g. page number, year

  • Reference 8 is indeed a journal article. We have updated it to include the proper format:

O’Grady, N. P., Alexander, E., Alazani, W., Alshamsi, F., Cuellar-Rodriguez, J., Jefferson, B. K., Kalil, A. C., Pastores, S. M., Patel, R., Van Duin, D., Weber, D. J., & Deresinski, S. (2023). Society of Critical Care Medicine and the Infectious Diseases Society of America Guidelines for evaluating new fever in adult patients in the ICU. Critical Care Medicine, 51(11), 1570–1586.

Line 277-279: Is reference 13 a journal article? If yes, please follow the proper format e.g. page number

  • We thank the reviewer for the comment. Reference 13 has been removed from the article as it was not essential to the content

Reviewer 3 Report

Comments and Suggestions for Authors

The manuscript by Yaniv Cojocaru and co-authors titled "The Utility of Blood Cultures in Non-Febrile Patients and Patients with Antibiotics Therapy in Internal Medicine Departments" is dedicated to comparing two groups of patients with fever — in the first group, blood cultures were taken within the first 12 hours after the onset of fever, while in the second group, cultures were taken after 12 hours. The authors found statistically significant differences in the microbiological yield between these two groups. A complex statistical analysis method was employed using two logistic regression models. Of course, I would have liked a more detailed description in the Materials and Methods section.

Other issues include:

Line 5: Remove designations like MD and PhD. Please add * for the corresponding author.

Lines 79-83: Latin names of microorganisms should be italicized. This also applies to lines 135-138.

Line 84: Why are the references not in the order of mention? Reference 7 appears before 14. Similarly, after reference 9 (line 198), reference 12 (line 202) follows immediately.

Table 3: Add * for COPD and AIDS.

Line 160: Remove CVA — it is not in Table 4.

Table 5: Add * for CVA and COPD.

Line 168: Suppl. table 1 should be provided in a separate file.

Lines 241-249: The list of abbreviations can be omitted; this is already present in the text.

Author Response

Response to reviewer 3:

Line 5: Remove designations like MD and PhD. Please add * for the corresponding author.

  • We have removed the designations (MD and PhD) as requested and added * to indicate the corresponding author and now it will appear as: Yaniv Cojocaru, 1, Lior Hassan 1, Lior Nesher 2, Tali Shafat 1,2 and Victor Novack* 1

Lines 79-83: Latin names of microorganisms should be italicized. This also applies to lines 135-138.

  • We have italicized the Latin names of microorganisms in lines 79-83 and 135-138 as requested: aureus, Streptococcus pneumoniae, Group A Streptococcus (GAS), Enterobacteriaceae, Haemophilus influenza, Pseudomonas aeruginosa, Bacteroidaceae, and Candida species etc…; the second, positive blood cultures with microorganisms associated with contamination: coagulase-negative staphylococci, Corynebacterium species, Cutibacterium, Bacillus species, and Micrococcus species

Line 84: Why are the references not in the order of mention? Reference 7 appears before 14. Similarly, after reference 9 (line 198), reference 12 (line 202) follows immediately.

  • We have reorganized the references to ensure they follow the order of mention in the text. New reference list:
  1. Tabriz MS, Riederer K, Baran J, Jr., Khatib R. Repeating blood cultures during hospital stay: practice pattern at a teaching hospital and a proposal for guidelines. Clin Microbiol Infect 2004; 10(7): 624-7
  2. Thomson, R. B., C. Corbin, and J. S. Tan. 1989. Timing of blood culture collection from febrile patients, abstr. C-227, p. 431. Abstr. 89th Annu. Meet. Am. Soc. Microbiol. 1989. American Society for Microbiology, Washington, DC
  3. Riedel, S., Bourbeau, P., Swartz, B., Brecher, S., Carroll, K. C., Stamper, P. D., Dunne, W. M., McCardle, T., Walk, N., Fiebelkorn, K., Sewell, D., Richter, S. S., Beekmann, S., & Doern, G. v. (2008). Timing of specimen collection for blood cultures from febrile patients with bacteremia. Journal of Clinical Microbiology, 46(4), 1381–1385.6
  4. Kee, P. P. L., Chinnappan, M., Nair, A., Yeak, D., Chen, A., Starr, M., Daley, A. J., Cheng, A. C., & Burgner, D. (2016). Diagnostic yield of timing blood culture collection relative to fever. Pediatric Infectious Disease Journal, 35(8), 846–8.7
  5. Lee A, Merrett S, Reller LB, Weinstein MP. Detection of bloodstream infections in adults: how many blood cultures are needed? J Clin Microbiol 2007; 45(11): 3546-8.
  6. Miller JM, Binnicker MJ, Campbell S, et al. A Guide to Utilization of the Microbiology Laboratory for Diagnosis of Infectious Diseases: 2018 Update by the Infectious Diseases Society of America and the American Society for Microbiology. Clin Infect Dis 2018. 2
  7. Neves L, Marra AR, Camargo TZ, et al. Correlation between mass and volume of collected blood with positivity of blood cultures. BMC Res Notes 2015; 8: 383. 3
  8. [dataset]NHSN Terminology | NHSN | CDC. (2023, December 20). https://www.cdc.gov/nhsn/cdaportal/terminology/index.html
  9. Walker, G. A., Runde, D., Rolston, D. M., Wiener, D., & Lee, J. (2013). Emergency department rectal temperatures in over 10 years: A retrospective observational study. World journal of emergency medicine, 4(2), 107–112.
  10. Kresovich-Wendler, K., Levitt, M. A., & Yearly, L. (1989). An evaluation of clinical predictors to determine need for rectal temperature measurement in the emergency department. The American journal of emergency medicine, 7(4), 391–394.
  11. O’Grady, N. P., Alexander, E., Alhazzani, W., Alshamsi, F., Cuellar-Rodriguez, J., Jefferson, B. K., Kalil, A. C., Pastores, S. M., Patel, R., Van Duin, D., Weber, D. J., & Deresinski, S. (2023). Society of Critical Care Medicine and the Infectious Diseases Society of America Guidelines for evaluating new fever in adult patients in the ICU. Critical Care Medicine, 51(11), 1570–1586. https://doi.org/10.1097/ccm.0000000000006022
  12. Weinstein, M. P. (1996). Current Blood Culture Methods and Systems: Clinical Concepts, Technology, and Interpretation of Results. In Clinical Infectious Diseases (1), 40–46.
  13. Michael L Wilson, Detection of bacteremia: Blood cultures and other diagnostic tests. In: UpToDate, Connor RF (Ed), Wolters Kluwer. Accessed Jun 13, 2024.
  14. Coburn, B., Morris, A. M., Tomlinson, G., & Detsky, A. S. (2012). Does this adult patient with suspected bacteremia require blood cultures? JAMA - Journal of the American Medical Association, 308(5), 502–511
  15. Jones GR, Lowes JA. The systemic inflammatory response syndrome as a predictor of bacteraemia and outcome from sepsis. QJM. 1996;89(7):515-522
  16. Shapiro NI,Wolfe RE,Wright SB, Moore R, Bates DW. Who needs a blood culture? prospectively de- rived and validated prediction rule. J EmergMed. 2008; 35(3):255-264.
  17. Leekha, S., Terrell, C. L., & Edson, R. S. (2011). General principles of antimicrobial therapy. Mayo Clinic proceedings, 86(2), 156–167.
  18. Lamy, B., Sundqvist, M., Idelevich, E. A., & ESCMID Study Group for Bloodstream Infections, Endocarditis and Sepsis (ESGBIES) (2020). Bloodstream infections - Standard and progress in pathogen diagnostics. Clinical microbiology and infection : the official publication of the European Society of Clinical Microbiology and Infectious Diseases, 26(2), 142–150
  19. Paul, M., Shani, V., Muchtar, E., Kariv, G., Robenshtok, E., & Leibovici, L. (2010). Systematic review and meta-analysis of the efficacy of appropriate empiric antibiotic therapy for sepsis. Antimicrobial agents and chemotherapy, 54(11), 4851–4863.

Table 3: Add * for COPD and AIDS.

  • We have added * for COPD and AIDS in Table 3 as requested:

Table 3. Factors Associated with Optimal Timing of Blood Culture Collection.

Characteristic

Odds Ratio

95% CI for Odds Ratio

P-value

Lower

Upper

Age(years)

0.99

0.99

0.99

<0.001

Pulse(bpm), per 10 bpm

1.28

1.27

1.30

<0.001

Systolic Blood Pressure(mmHg), per 10 mmHg

1.06

1.05

1.07

<0.001

C-reactive protein (mg/dl) per 5 mg/dL

1.10

1.09

1.11

<0.001

White blood cells (106/ml)

0.99

0.98

0.99

<0.001

Congestive heart failure

0.86

0.81

0.91

<0.001

Dementia

1.23

1.14

1.34

<0.001

COPD*

0.85

0.81

0.90

<0.001

Rheumatological disease

0.85

0.76

0.94

0.003

Diabetes Mellitus with chronic complications

0.93

0.88

0.97

<0.05

Hemiplegia/Paraplegia

1.30

1.22

1.40

<0.001

Renal disease

0.79

0.75

0.83

<0.001

Malignancy

0.92

0.87

0.98

0.007

Moderate to severe liver disease

0.79

0.66

0.95

0.013

AIDS*

0.53

0.36

0.77

0.001

* COPD- Chronic Obstructive Pulmonary Disease, AIDS-acquired immunodeficiency syndrome,

Line 160: Remove CVA — it is not in Table 4.

  • We removed it:

Characteristic

Odds Ratio

95% CI for the odds ratio

P-value

Lower

Upper

Optimal timing

1.23

1.12

1.35

<0.001

Age (years)

1.01

1.00

1.01

0.001>

Pulse(bpm), per 10 bpm

1.05

1.04

1.07

0.001>

Systolic Blood Pressure(mmHg), per 10 mmHg

0.88

0.86

0.90

0.001>

C-reactive protein (mg/dl) per 5 mg/dL

1.06

1.04

1.07

0.043

White blood cells (106/ml)

1.01

1.00

1.01

0.001>

Diabetes Mellitus with chronic complications

1.38

1.22

1.44

0.001>

Renal disease

1.15

1.04

1.28

0.006

Moderate to severe liver disease

2.21

1.67

2.93

0.001

Any Malignancy

1.20

1.07

1.33

0.001>

AIDS

2.32

1.32

4.05

0.003

  • *AIDS-acquired immunodeficiency syndrome.

Table 5: Add * for CVA and COPD.

  • We added those terms to table 5:

Characteristic

Odds ratio

95% CI for the odds ratio

P-value

Lower

Upper

Optimal timing

1.17

1.03

1.34

0.018

Age (years)

1.01

1.00

1.01

0.001>

Gender-male

0.86

0.75

0.98

0.024

Systolic Blood Pressure(mmHg), per 10 mmHg

0.95

0.92

0.98

<0.001

C-reactive protein (mg/dl) per 5 mg/dL

0.96

0.93

0.99

0.024

CVA*

1.25

1.07

1.45

0.004

Dementia

0.76

0.58

0.99

0.041

COPD*

0.79

0.68

0.93

0.003

Renal disease

0.79

0.68

0.93

0.004

Rheumatological disease

0.48

0.30

0.79

0.003

* CVA – cerebrovascular accident, COPD- Chronic obstructive pulmonary disease.

Line 168: Suppl. table 1 should be provided in a separate file.

  • We have provided Suppl. Table 1 in a separate file as requested:

Characteristics

True bacteremia, n = 4,045(5.5)

Contamination and no growth, n=69,745(94.5%)

SMD

Optimal timing

1,562(38.6%)

24,054(34.5%)

0.086

Age(y.)

66.13±17.41

65.25±18.44

0.049

    Male, n (%)

2,454(60.7%)

40,950(58.7%)

Comorbidities

   CHF, n (%)

1,047(25.9%)

15,621(22.4%)

0.082

   Stroke, n (%)

1,020(25.2%)

17,172(24.6%)

0.014

   Dementia, n (%)

308(7.6%)

4557(6.5%)

0.042

   Chronic pulmonary disease, n (%)

1,111(27.5%)

18,611(26.7%)

0.018

  Rheumatological disease, n (%)

135(3.3%)

2,723(3.9%)

0.030

   Diabetes mellitus, n (%)

1,740(43.0%)

27,124(38.9%)

0.084

   Diabetes mellitus with chronic complications, n (%)

684(16.9%)

10,177(14.6%)

0.064

   Peripheral Vascular disease, n (%)

728(18.0%)

10,632(15.2%)

0.074

    Malignancy, n (%)

835(20.6%)

13,298(19.1%)

0.040

    Renal disease, n (%)

1,285(31.8%)

18,847(27.0%)

0.104

Vital signs

  Pulse (bpm)

99.0±39.6

94.1±21.8

0.154

  Systolic Blood pressure(mmHg)

127.3±23.8

133.2±22.7

0.258

Blood tests

  White blood cells (mm3/mL)

11.83±8.75

11.8±8.7

0.240

  C-reactive protein (mg/dL)

15.29±13.78

12.72±11.19

0.239

Lines 241-249: The list of abbreviations can be omitted; this is already present in the text

response: We have omitted the list of abbreviations

Round 2

Reviewer 2 Report

Comments and Suggestions for Authors

Response to reviewer 2:

Third Review

Some of the methodology is a bit ambiguous. For instance, what if coagulase-negative staph and “contaminants” show up in two sets of blood culture of the same patient – would that be considered contaminants? What about some other significant pathogens not included in this study list of pathogens (e.g. Viridans group streptococcus, obligate anaerobes other than Bacteroidaceae). Without knowing this detail, it is hard to determine whether this study had robust methodology.

  • We agree with the reviewer. Therefore, the methods section contains the following explanation: " We stratified the blood culture results into three groups. The first group included true-positive blood cultures with clinically significant microorganisms, such as Staphylococcus aureus, Streptococcus pneumoniae, Group A Streptococcus (GAS), Enterobacteriaceae, Haemophilus influenzae, Pseudomonas aeruginosa, Bacteroidaceae, and Candida species. The second group consisted of positive blood cultures containing microorganisms commonly associated with contamination, including coagulase-negative staphylococci, Corynebacterium species, Cutibacterium, Bacillus species, and Micrococcus species. However, some of these contaminants may act as potential pathogens.When pathogens typically associated with contamination were present in more than two sets of blood cultures, we excluded them from the analysis, as they might indicate a clinical syndrome requiring follow-up with repeated blood culture testing.
  • Furthermore, we have updated the limitation section to state: "The exclusion of the patients with suspected contaminant found in more than one blood culture could potentially lead to a misclassification bias."
  • The authors’ response stated “we stratified the blood culture results into three groups.” The authors described the first and second groups here. What about the third group?

 The study define fever as per rectum temperature of 38.3 degrees Celsius or higher. But it is unclear what about measurement of fever by other means as such as forehead and axillary thermometers and measurement taken prior to arrival at the hospital. These patients would be classified to the group with “suboptimal blood culture timing.”

  • We agree with the reviewer comment regarding other means of measuring temperature and possible pre-hospital setting recorded temperature. However, per our study design we have classified the patient population based only on the oral or rectal temperature taken in the hospital close to the drawing of the blood cultures. The preadmission temperature was not recorded.
  • Then the revised manuscript should have mentioned about oral temperature. I did a search using terms like “oral” and “mouth” in the manuscript and cannot find any.

Although the authors show there were some clinical parameters leading to optimal blood culture timing, such as systolic blood pressure and pulse (Table 3). Many hospitals nowadays would have a sepsis protocol that advise blood culture taken when there are signs of fever with unstable vital signs. This manuscript needs better justifications on why the current study findings are novel.

  • We believe that the findings of the current study stress the need to relay on the holistic view of the signs and symptoms (and in a near future a personalized medicine approach), rather than follow strict protocols. We have shown that while the presence of fever and no prior antibiotics increase the chance of yielding a true positive culture, testing the blood cultures in patients with no fever and/or prior antibiotics therapy did result in the non-negligible rate of true bacteremia. Moreover, we have identified factors associated with the increase in the bacteriological yield that following the prospective validation may be used as a basis for the creation of the personalized testing protocols
  • Then, are the authors saying this non-negligible rate of true bacteremia is good or bad? Their conclusion in abstract seems to be focused on suboptimal timing of blood cultures taken, not about non-negligible rate of true bacteremia.
  • If the point of this study is to create personalized testing protocols, then how is it different from the sepsis protocols that are already in place in guidelines? What is novel about this study?

Second Review

General comments

Generally speaking, the journal editors would require the manuscript authors to address point-by-point of the reviewers’ comments to ensure things are not missed. Some of my previous comments are not addressed in the authors’ current reply. I will let the editors to decide whether my previous reviewer comments still need to be addressed. For now, I will respond to the comments raised in the authors’ current reply.

“the temperature measurement method, in our institution temperature is measured through the mouth or the rectum and so far, we have referred in our study to oral or rectal temperature only.”

  • Then the revised manuscript should have mentioned about oral temperature. I did a search using terms like “oral” and “mouth” in the manuscript and cannot find any.

“During that period, a lack of manpower resulted in delays of up to 12 hours between the collection of blood cultures and their arrival at the bacteriological laboratory for processing. Over time, this issue has been addressed through improvements in logistical support, particularly the availability of a dedicated transport team to ensure timely delivery of blood cultures to the laboratory. Consequently, we have reached a consensus that a 12-hour time window, both before and after collection, is reasonable and acceptable.”

  • Thanks for the explanation. But that still does not explain why the study cannot have 2 subgroups: 1) delay over 12 hours and 2) prior antibiotics. It seems to me these two subgroups would have very different contexts and should not be mixed.

“Indeed, you’re right, fever alone is poor predictor of bacteremia that is why we choose to add antibiotics as part of optimal or non optimal condition.”

  • As mentioned above, 1) delay over 12 hours and 2) prior antibiotics should be 2 separate groups.

“For this reason, we decided not to consider these self-reported measurements and instead focus exclusively on the data objectively collected within the hospital setting.”

  • Then, it means many of this “suboptimal blood culture collection” as defined in this study is not really suboptimal in a clinical sense. A patient with subjective history of fever and systemic illness but no clear diagnosis does warrant some investigations, such as blood culture. Then, perhaps, there is nothing wrong with the rationale of some of the physicians’ sampling decisions.

“Our aim was not to determine the sensitivity and specificity or pre-test probability of bacteremia. The aim of our work was to find other factors that can contribute to the decision to which patient blood cultures will have highest chances for finding bacteremia under the optimal timing conditions.” 

  • But your conclusion still states “fever onset alone probably lacks both sensitivity and specificity in guiding physician decisions to obtain blood cultures. Instead, the decision should be informed by the pre-test probability of bacteremia, considering clinical suspicion and the patient's underlying diseases.”
  • Your study does not show sensitivity, specificity and pre-test probability. It is best to avoid using these terminologies in your conclusion.

Author Response

Thank you for your review. Below please find point-by-point response and description of the implemented changes

General comments

Comment 1:Some of the methodology is a bit ambiguous. For instance, what if coagulase-negative staph and “contaminants” show up in two sets of blood culture of the same patient – would that be considered contaminants? What about some other significant pathogens not included in this study list of pathogens (e.g. Viridans group streptococcus, obligate anaerobes other than Bacteroidaceae). Without knowing this detail, it is hard to determine whether this study had robust methodology.

Response 1: We agree with the reviewer. Therefore, the methods section contains the following explanation: " We stratified the blood culture results into three groups. The first group included true-positive blood cultures with clinically significant microorganisms, such as Staphylococcus aureus, Streptococcus pneumoniae, Group A Streptococcus (GAS), Enterobacteriaceae, Haemophilus influenzae, Pseudomonas aeruginosa, Bacteroidaceae, and Candida species. The second group comprised patients with positive blood cultures d/t microorganisms commonly associated with contamination, including coagulase-negative staphylococci, Corynebacterium species, Cutibacterium, Bacillus species, and Micrococcus species.” Indeed, some of these contaminants may act as potential pathogens. When pathogens typically associated with contamination were present in more than two sets of blood cultures, we excluded them from the analysis, as they might indicate a clinical syndrome requiring follow-up with repeated blood culture testing.

Furthermore, we have updated the limitation section to state: "The exclusion of the patients with suspected contaminant found in more than one blood culture could potentially lead to a misclassification bias, yet the low number of such cases (less than 20) reassures that the study results are robust."

Comment 2:The authors’ response stated “we stratified the blood culture results into three groups.” The authors described the first and second groups here. What about the third group?

Response 2 : We agree with the reviewer and we add to the text the following sentence " The third group the blood cultures with no growth.

Comment 3:“The temperature measurement method, in our institution temperature is measured through the mouth or the rectum and so far, we have referred in our study to oral or rectal temperature only.”

Then the revised manuscript should have mentioned about oral temperature. I did a search using terms like “oral” and “mouth” in the manuscript and cannot find any.

Response 3: We thank the reviewer for the comment. The definition of fever can be found in our manuscript in lines 112-117 : "Optimal timing for blood culture collection was defined based on two specific criteria: documented fever and the absence of prior antibiotic administration. Fever was identified using one of the two validated methods employed at our institution—rectal or oral temperature measurement—with a threshold of ≥38.3°C recorded within 12 hours before or after blood culture collection, accounting for potential delays in sample transport to the microbiology laboratory."

Comment 4: We believe that the findings of the current study stress the need to relay on the holistic view of the signs and symptoms (and in a near future a personalized medicine approach), rather than follow strict protocols. We have shown that while the presence of fever and no prior antibiotics increase the chance of yielding a true positive culture, testing the blood cultures in patients with no fever and/or prior antibiotics therapy did result in the non-negligible rate of true bacteremia. Moreover, we have identified factors associated with the increase in the bacteriological yield that following the prospective validation may be used as a basis for the creation of the personalized testing protocols"

Then, are the authors saying this non-negligible rate of true bacteremia is good or bad? Their conclusion in abstract seems to be focused on suboptimal timing of blood cultures taken, not about non-negligible rate of true bacteremia.

Response 4: We appreciate the reviewer's insight. A non-negligible rate of true bacteremia in patients without fever and/or prior antibiotic treatment is neither inherently good nor bad.  It shows however, that even in “non-optimal” clinical situation, the yield is sufficient to warrant the change in the clinical paradigm (“take cultures when the patient has fever”). We have updated the conclusion in the abstract: "Nearly two-thirds of patients hospitalized due to suspected infection did not have their blood cultures taken at the optimal time. These cultures resulted in slightly lower, albeit non-negligible rate of positive bacteriological yield. Our findings underscore the importance of integrating clinical judgment, patient-specific risk factors, and evidence-based criteria when deciding to perform blood cultures, rather than relying solely on fever as an indicator."

Comment 5: If the point of this study is to create personalized testing protocols, then how is it different from the sepsis protocols that are already in place in guidelines? What is novel about this study?

Response 5: Indeed, sepsis protocols play a crucial role in evaluating septic patients. However, as stated in UpToDate, routine blood cultures are warranted only for patients who exhibit symptoms, radiographic evidence, or laboratory test results suggesting the presence of syndromes associated with a high likelihood of bacteremia. Our study is novel because, while protocols are important, they often do not take into account that each patient is unique. Moreover, we have shown that the prevalent clinical gestalt focusing on fever as a trigger for blood culturing is not entirely accurate. Personalized testing protocols should consider additional factors beyond standardized guidelines to ensure optimal patient care.

Comment 6:“During that period, a lack of manpower resulted in delays of up to 12 hours between the collection of blood cultures and their arrival at the bacteriological laboratory for processing. Over time, this issue has been addressed through improvements in logistical support, particularly the availability of a dedicated transport team to ensure timely delivery of blood cultures to the laboratory. Consequently, we have reached a consensus that a 12-hour time window, both before and after collection, is reasonable and acceptable.”

Thanks for the explanation. But that still does not explain why the study cannot have 2 subgroups: 1) delay over 12 hours and 2) prior antibiotics. It seems to me these two subgroups would have very different contexts and should not be mixed.

“Indeed, you’re right, fever alone is poor predictor of bacteremia that is why we choose to add antibiotics as part of optimal or non-optimal condition.”

As mentioned above, 1) delay over 12 hours and 2) prior antibiotics should be 2 separate groups."

Response 6: We appreciate the reviewer's comment about separating the analysis into two subgroups. To directly address this suggestion, we performed a sensitivity analysis, introducing both variables separately into the model.  This new analysis showed that each exposure independently affected the likelihood of obtaining a true positive culture. The absence of prior antibiotic therapy had an OR of 1.86 (95% CI 1.45–2.40), and obtaining cultures within 12 hours had an OR of 1.26 (95% CI 1.15–1.38). The interaction between these variables was not significant. These results have now been included in the Results section lines 192-127 of the manuscript, and we have also expanded our discussion to address these findings and their implications. 

Comment 7: “For this reason, we decided not to consider these self-reported measurements and instead focus exclusively on the data objectively collected within the hospital setting.”

Then, it means many of this “suboptimal blood culture collection” as defined in this study is not really suboptimal in a clinical sense. A patient with subjective history of fever and systemic illness but no clear diagnosis does warrant some investigations, such as blood culture. Then, perhaps, there is nothing wrong with the rationale of some of the physicians’ sampling decisions.

Response 7: Precisely! We agree with the reviewer. There is nothing wrong with the rationale of some of the physicians' sampling decisions to obtain blood cultures in patients with subjective fever and systemic illness with no clear diagnosis which justify our claim that "the findings of the current study stress the need to relay on the holistic view of the signs and symptoms (and in a near future a personalized medicine approach), rather than follow strict protocols."  And” " In conclusion, our findings underscore the importance of integrating clinical judgment, patient-specific risk factors, and evidence-based criteria when deciding to perform blood cultures, rather than relying solely on fever as an indicator.”

Comment 8 : “Our aim was not to determine the sensitivity and specificity or pre-test probability of bacteremia. The aim of our work was to find other factors that can contribute to the decision to which patient blood cultures will have highest chances for finding bacteremia under the optimal timing conditions.” 

But your conclusion still states “fever onset alone probably lacks both sensitivity and specificity in guiding physician decisions to obtain blood cultures. Instead, the decision should be informed by the pre-test probability of bacteremia, considering clinical suspicion and the patient's underlying diseases". Your study does not show sensitivity, specificity and pre-test probability. It is best to avoid using these terminologies in your conclusion.

Response 9: We would like to express our concern to the reviewer regarding possible misunderstanding. In our updated manuscript sent to the editors we followed the reviewer recommendation : " In conclusion, our findings underscore the importance of integrating clinical judgment, patient-specific risk factors, and evidence-based criteria when deciding to perform blood cultures, rather than relying solely on fever as an indicator. This approach may improve diagnostic accuracy and support more effective and targeted patient management, aligning with the principles of antibiotic stewardship and resource optimization."